# Optimizing the Clinical Use of High-Sensitivity Troponin Assays: A Review

**DOI:** 10.3390/diagnostics14010087

**Published:** 2023-12-30

**Authors:** Dipti Tiwari, Tar Choon Aw

**Affiliations:** 1Independent Researcher, Singapore 069046, Singapore; tiwari.dipti02@gmail.com; 2Department of Laboratory Medicine, Changi General Hospital, Singapore 529889, Singapore; 3Yong Loo Lin School of Medicine, National University of Singapore (NUS), Singapore 119228, Singapore; 4Pathology Academic Clinical Program, Duke-NUS Graduate School of Medicine, Singapore 169857, Singapore

**Keywords:** acute myocardial infarction, cardiac biomarkers, high-sensitivity cardiac troponins, serial troponin testing

## Abstract

Ischemic heart diseases (IHDs) remain a global health concern. Many IHD cases go undiagnosed due to challenges in the initial diagnostic process, particularly in cases of acute myocardial infarction (AMI). High-sensitivity cardiac troponin (hs-cTn) assays have revolutionized myocardial injury assessment, but variations in diagnostic cut-off values and population differences have raised challenges. This review addresses essential laboratory and clinical considerations for hs-cTn assays. Laboratory guidelines discuss the importance of establishing standardized 99th-percentile upper reference limits (URLs) considering factors such as age, sex, health status, and analytical precision. The reference population should exclude individuals with comorbidities like diabetes and renal disease, and rigorous selection is crucial. Some clinical guidelines emphasize the significance of sex-specific URL limits while others do not. They highlight the use of serial troponin assays for AMI diagnosis. In addition, timely reporting of accurate hs-cTn results is essential for effective clinical use. This review aims to provide a clearer understanding among laboratory professionals and clinicians on how to optimize the use of hs-cTn assays in clinical settings in order to ensure accurate AMI diagnosis and thus improve patient care and outcomes.

## 1. Introduction

Cardiovascular diseases are the leading cause of death globally. In Singapore, ischemic heart diseases (IHDs) accounted for 19.7% of total deaths in 2022 [1]. A significant number of patients with acute coronary syndrome (ACS) often fail to receive timely treatment owing to incorrect preliminary diagnosis [2]. Most cases of acute myocardial infarction (AMI) present with chest pain, and a proper, rapid diagnostic assessment is essential so that early and effective treatment can be implemented. Similarly, prompt the rule-out of an AMI would avoid unnecessary hospital admissions.

In the initial evaluation of suspected ACS clinical assessment, 12-lead electrocardiogram (ECG), and cardiac troponins (cTn) play a crucial role [3,4]. ST-elevation MI (STEMI) requires immediate reperfusion therapy while very high-risk non-STEMI needs immediate invasive strategy (angiography ± percutaneous coronary intervention). For non-STEMI with high-risk features, angiography within 24 h can be considered. This algorithm is illustrated in Figure 1. Over the recent years, high-sensitivity cardiac troponin (hs-cTn) assays have revolutionized the assessment of myocardial stress and injury due to their superior sensitivity and accuracy in detecting even minor cardiac damage [5]. Patients with STEMI or very high-risk non-STEMI do not need troponin for diagnosis, but their management should not be delayed by waiting for troponin results to be available. However, hs-cTn is often included in the initial blood draw for baseline assessment in such patients and has prognostic value in predicting in-hospital and long-term mortality. For the remining patients with suspected ACS, hs-cTn is pivotal for confirmation of diagnosis, risk assessment/prognostication and subsequent management. 

Substantial variations exist between diagnostic cut-off hs-cTn values across individual laboratories, which can be attributed to biological as well as analytical variabilities. With improvements in hs-cTn analytical assays, population differences in troponin levels have also been revealed, which could confound clinical decision making [6]. However, these differences are more apparent than real due to differences in the composition of the reference population and the statistical treatment of the reference data [7].

In this review, we highlight the essential considerations (both laboratory and clinical) for the use of hs-cTn assays. Recommendations for establishing reference limits are discussed, considering population variances, particularly in the context of assessing chest pain to confirm or exclude AMI. We attempt to foster a cohesive understanding among both laboratory professionals and clinicians regarding the optimal utilization of hs-cTn assays in clinical settings and highlight some challenges.

## 2. Laboratory Considerations for hs-cTn

In the recent past, it has become evident that the hs-cTn 99th-percentile upper reference limit (URL) values are influenced by factors such as age [8,9,10], sex [4,11], health status [12,13], and the statistical approach used to derive the URL [14]. In light of this, reputable scientific societies such as the International Federation of Clinical Chemistry and Laboratory Medicine (IFCC) have issued specific recommendations aimed at standardizing the establishment of 99th-percentile URLs for hs-cTn [15]. These recommendations are intended to promote consistency in the interpretation and application of hs-cTn.

### 2.1. Analytical Designation

To be classified as a high-sensitivity troponin assay, it must meet the following two analytical criteria: (1) have a total imprecision (CV) of ≤10% at the sex-specific 99th-percentile levels, and (2) detect troponin concentrations above the assay’s limit of detection (LoD) in at least 50% of healthy individuals for both men and women [14]. Here, it would be useful to review the meaning of the limit of blank (LoB), LoD, and the limit of quantitation (LoQ) of troponin assays. The CLSI guideline EP17-A2 provides recommendations to standardize the related terminology and statistically define the detection capabilities in terms of the LoB, LoD, and LoQ [16]. The LoB refers to the highest troponin signal generated by the assay while analyzing replicates of a sample containing no analyte (zero calibrator) for the cardiac troponin assay. It is derived statistically as LoB = mean (zero calibrator) + 1.645 × SD (zero calibrator) [17]. Thus, it represents the ‘signal noise’ inherent in the analytical system. The LoD, which always exceeds the LoB, represents the lowest concentration of cardiac troponin that the assay can reliably detect with a 95% confidence level but this may not be precise enough for accurate reporting. The LoQ is the lowest cTn value that demonstrates a 20% imprecision (CV) and can be reliably reported; the LoQ is always greater than the LoD [18].

Various platforms have reported significantly lower hs-cTn values among women compared with men, which may be attributed to their lower cardiac mass [19]. A recent cardiac magnetic resonance imaging (MRI) study reported a left ventricular mass of 39 g/m^2^ for women and 50 g/m^2^ for men [20]. Thus, the use of a single diagnostic threshold for cTn values in both men and women may result in the underdiagnosis of myocardial infarction in women and overdiagnosis in men [21,22,23]. This has led to the inclusion of sex-specific 99th-percentile URLs for hs-cTn in the universal definitions of myocardial infarction [4] and in the International Federation of Clinical Chemistry (IFCC) guidelines [24,25]. However, the recent European guidelines continue to advocate for single gender-independent troponin cut-offs [26]. 

### 2.2. Defining the Reference Population

The reference cohort should comprise healthy individuals; preinclusion screening should be performed through a questionnaire survey and clinical examination to ensure that the population is healthy [27]. Certain comorbidities have been reported to substantially influence the measured hs-cTn concentrations, ultimately affecting the determination of URLs and rendering it unacceptable to include individuals with such comorbidities in the reference cohort. Subclinical myocardial injury and hemodynamic stress raise serum levels of troponin [28]. Furthermore, elevated hs-cTnT levels have been associated with increased incidence rates of cardiac failure [29]. As such, it has been recommended to exclude individuals with N terminal pro-brain natriuretic peptide (NT-proBNP) levels >125 ng/L or brain natriuretic peptide (BNP) levels >35 ng/L [30]. Diabetes mellitus and renal dysfunction are other conditions that need to be excluded. Individuals with chronic hyperglycemia exhibit reduced troponin elimination as a consequence of reduced glomerular filtration. Additionally, microvascular damage due to hyperglycemia may lead to myocardial injury and ischemia, subsequently increasing the levels of cTns in the circulation. Thus, glycosylated hemoglobin (HbA1c) screening would be useful while selecting the reference population; individuals with HbA1c values ≥48 mmol/mol (6.5%) are excluded [24,31].

Another important condition that needs to be excluded is chronic renal disease. Higher troponin concentrations have been reported in the presence of renal dysfunction [32,33], which can be partially explained by impaired renal clearance [34]; this is especially so with troponin-T compared to Troponin I [35]. Studies have also reported that troponin increases with a declining estimated glomerular filtration rate (eGFR) between 60 and 90 mL/min/1.73 m^2^ [36]. In a prospective study including cardio-renal healthy participants in Singapore, it was observed that for every 10 mL/min/1.73 m^2^ reduction in eGFR below 90 mL/min/1.73 m^2^, there was a significant stepwise increase in hs-cTnT concentrations, though not for hs-cTnI [20]. Thus, we feel that the more liberal eGFR recommendation by the expert committee guideline should be tightened to include only subjects with eGFR >90 mL/min/1.73 m^2^. The current IFCC guidelines recommend excluding only those with eGFR <60 mL/min/1.73 m^2^ [24].

Of note, several studies [37,38] have highlighted the influence of the reference population selection strategy on the hs-cTn 99th-percentile URLs. Reference cohorts that rigorously exclude individuals taking medications associated with cardiovascular disease or risk factors (e.g., antihypertensive, antidiabetes, or lipid-lowering drugs) tend to exhibit lower URLs compared to populations with less stringent screening criteria. The Association for Diagnostics and Laboratory Medicine (ADLM; formerly AACC) [39] also recommends the exclusion of those with cancer and thyroid diseases. Individuals with an abnormal body mass index (BMI) may also need to be excluded as obesity has been found to be independently associated with elevated hs-cTn levels [40]. 

### 2.3. Derivation of the 99th Percentile

The reference population should ideally consist of approximately equal proportions of 400 males and females each [24,39], spanning an age range from 18 to 80 years, with all relevant ethnic and racial groups adequately represented. As opposed to the earlier recommendation of including a minimum of 300 men and 300 women, the recent report of the IFCC Task Force on Clinical Applications of Cardiac Biomarkers (IFCC TF-CB) now recommends a minimum of 800 total subjects (400 men and 400 women) to be included in the reference cohort [24]. A sample size of 300 subjects per gender was sufficient to generate a URL with only a 90% confidence interval (CI) ±10% while a sample size of 400 subjects per gender provides a URL with a 95% CI ± 10%.

It should be noted that the sample size requirement is also influenced by the statistical method used to calculate the URL. While various statistical methods can be employed to establish reference limits, the Clinical and Laboratory Standards Institute (CLSI) suggests using a one-tailed nonparametric method [41]. Given the meticulous selection of participants in the healthy cohort, the risk of outliers arising from undetected subclinical comorbidities should be minimized. Therefore, the process of excluding outliers should also be thorough. The Reed/Dixon criteria is a preferable approach by some labs, as it tends to exclude fewer subjects compared to the Tukey method [42]; we favor the more stringent Tukey approach.

The 99th-percentile URL is a key metric that is meticulously derived since the URL is used to minimize any statistical uncertainty. It is important to note that the 99th-percentile URL is not a magic number but simply a value that is used to define the upper limit of the normal range for hs-cTn in the reference population. Values above the URL are considered to be abnormal and may indicate the presence of myocardial injury or infarction; the higher the value, the greater the likelihood is of AMI. However, for interpreting hs-cTn results, it is important to consider some analytical factors as well as clinical conditions that may lead to altered troponin levels. Besides MI, there are many other non-MI clinical entities associated with elevated troponins that should be recognized while interpreting hs-cTn results (Table 1). These conditions may be due to impaired oxygen supply and demand as well as a complex mix of comorbid conditions acting in concert [26]. 

### 2.4. Quality Control (QC) Recommendations

With the evolution of hs-cTn assays over the recent years and the pivotal role assumed by troponins in the diagnosis of AMI, the analytical performance criteria have become more rigorous for hs-cTn assays. It is of utmost importance to monitor assay performance across a range of concentrations that are relevant for clinical decision making, including the 99th-percentile URL. Thus, recent guidelines [52] have recommended the use of three levels of QC materials for hs-cTn assays, as follows:Level 1 QC: This concentration should fall within the range between the LoD and the lowest sex-specific 99th percentile. It is important to evaluate the lower end of the assay’s analytical sensitivity and precision, since low cTn levels (<LoD) are used to rule out AMI/myocardial injury.Level 2 QC: This concentration should be higher than but close to (within 20% of) the highest sex-specific 99th-percentile URL. It allows for the assessment of the assay’s accuracy and precision near the URL.Level 3 QC: This concentration should be significantly elevated to challenge the upper analytical range of reportable cTn results. It involves using concentrations that are multiples above the 99th-percentile concentration. This is important to assess the reproducibility of the assay at high cTn concentrations.

### 2.5. Reporting Units for hs-cTn

Besides the QC recommendations, the consensus among prominent specialist lab organizations like the IFCC and the ADLM is that hs-cTn values should be expressed in nanograms per liter (ng/L) and reported as whole numbers [24,39]. This avoids confusion and potential transcription errors, especially when comparing results with non-high-sensitivity assays. This scenario occurs when the Emergency Department deploys point-of-care (POC) troponin testing (conventional cTn) while the central laboratory uses a hs-cTn method. In addition, adopting this reporting convention helps avoid errors in data recording and ensures clarity and consistency in the reporting of hs-cTn results.

## 3. Clinical Considerations for hs-cTn to Rule In and Rule Out AMI

Cardiac troponins (cTnT and cTnI) are proteins that regulate calcium-mediated interactions between actin and myosin; their serum levels are elevated in the presence of damaged heart muscle cells and are associated with increased morbidity and mortality [53,54]. Hs-cTn assays are more sensitive than traditional cardiac troponin assays, meaning that they can detect smaller amounts of cardiac troponin in the blood. This makes hs-cTn assays better at diagnosing AMI earlier. 

In all patients with suspected ACS, hs-cTn is recommended [26]. The diagnosis of AMI is based on a combination of clinical factors, such as the patient’s symptoms and medical history, and laboratory results, such as the hs-cTn levels. The Fourth Universal Definition of Myocardial Infarction [4] defines AMI as a rise and/or fall in cardiac troponin with at least one value above the 99th-percentile URL and at least one of the following:Symptoms of myocardial ischemia;New or presumed new significant ST-segment or T-wave changes;Development of pathological Q waves on the ECG;Imaging evidence of new loss of viable myocardium;Identification of an intracoronary thrombus by angiography or autopsy in the setting of ischemia.

As mentioned earlier (Section 2.1), owing to the physiological differences between men and women [20,55], higher troponin concentrations are observed in males than in females. A previous study also demonstrated much lower hs-cTn levels in healthy younger women (aged 20–39 years) than in older subjects (30–39 years) and thereby recommended to exclude the former group from the reference population for the determination of 99th-percentile URLs [56]. In addition, the prevalence of AMI and myocardial injury in subjects < 40 years is very low. As such, several studies have also found an elevation in serum troponin concentration with age [57,58,59,60]. In a previous study [20] investigating normal hs-cTn values in a large (*n* = 779) multi-centric Asian population (age 17–88 years), detectable (>LoD) hs-cTnI and hs-cTnT concentrations were found in 38.4% and 29.7% of participants, respectively, which increased to 56.6% and 54.6%, respectively, in subjects aged above 60 years. Moreover, the sex-specific 99th-percentile URLs for men and women were found to be 38.8 and 14.4 ng/L, respectively, for hs-cTnI and 16.8 and 11.9 ng/L, respectively, for hs-cTnT [20]. These results highlight the importance of age- and sex-specific URL limits for clinical decision making [61]. In fact, the latest 2023 ESC guidelines on ACS management [26] highlights the difference in troponin levels among very young healthy subjects versus healthy very old subjects may vary by up to 300%, with gender contributing a 40% difference.

### 3.1. Cardiac Troponin Levels Correlate with the Severity of Myocardial Injury

Troponin levels beyond the 99th-percentile URL are indicative of myocardial injury, including AMI. Elevated levels of hs-cTn are linked to a greater burden of coronary atherosclerosis, an accelerated progression of coronary artery disease (CAD), and an increased risk for both all-cause mortality and the occurrence of cardiovascular events [62,63]. Data obtained from some large-scale studies further draw attention to the fact that the interpretation of hs-cTn concentrations as a continuum of quantitative values rather than a binary positive/negative improves their diagnostic value [64]. When assessing patients with acute chest pain, low elevations of hs-cTn levels (just above the 99th-percentile URL) are associated with a reduced likelihood of AMI and a higher negative predictive value (NPV). Such low hs-cTn elevations should alert the clinician to consider other entities in the differential diagnosis as the positive predictive value (PPV) for AMI is likely to be low. Conversely, higher hs-cTn levels indicate a greater likelihood of AMI and a higher PPV, leading to consideration of a narrower range of alternative diagnoses. Besides the presence or absence of MI, four main factors (age, renal dysfunction, time of chest pain onset and gender) can impact troponin levels [26].

### 3.2. Serial Troponin Assays to Rule In and Rule Out AMI

Guidelines from the American Heart Association (AHA) and European Society of Cardiology (ESC) recommend serial hs-cTn measurements (0/1 h, 0/2 h or 0/3 h) to rule in or rule out AMI [25,26], with the ESC preferring an earlier second blood draw. This is also endorsed by the Asia–Pacific Consensus Group [65,66]. Serial hs-cTn algorithms have been derived from large multi-site studies mostly employing hsTnT (Roche) and hsTnI (Abbott). The optimal hs-cTn cut-offs for rule-out (Roche < 5 ng/L, Abbott < 4 ng/L) were chosen to provide an NPV of >99% while that for rule-in (Roche 52 ng/L, Abbott 64 ng/L) were set to obtain an NPV of >70% [26]. Thus, patients can be categorized into three groups—rule-out, observation, and rule-in (illustrated in Figure 2 for a 0/2 h algorithm). A prior study involving individuals with suspected acute coronary syndrome demonstrated that those who consistently had two elevated hs-cTnT measurements 1–7 h apart had the highest mortality [67]. In addition, this group had a notably higher risk of short-term (but not long-term) mortality as the relative rise in troponin levels exceeded 20% between measurements. Conversely, individuals with two normal hs-cTnT values had a very low risk of death, and this risk was not affected by changes, whether relative or absolute, between the measurements. This study underscores the importance of serial troponin sampling in providing valuable prognostic information beyond solely diagnosing myocardial infarction.

At levels around the LoD, a single hs-cTn can be used to rule out AMI with a high degree of certainty (>99%) [68]. This single troponin measurement to rule out MI has been recently reviewed [69]. The authors emphasize that AMI can be safely excluded in hs-cTn around the LoD taken from patients with chest pain >2 h and a non-ischemic ECG. However, to rule in AMI, a single hs-cTn value above the 99th-percentile URL is not specific or sensitive enough as it may be indicative of subtle myocardial injury and not necessarily AMI. To improve the rule-in of AMI, hs-cTn levels >10 times the upper limit of normal have been suggested [65]. ESC serial testing algorithms are assay-specific and details of the different cut-offs for rule-out, rule-in and the significant difference (deltas) between the initial and second hs-cTn values for several analytical platforms besides Roche and Abbott are available [26]. 

In the real world of busy emergency departments, a 0/1 h protocol would be hard to achieve. In addition, very many laboratories are unable to even meet the 60 min turn-around time from sample receipt in the laboratory to results reporting. Thus a 0/2 h strategy would be more achievable. In fact, the Asia–Pacific Consensus group recommends a 0/3 h serial troponin testing for hs-TnI because of logistical constraints [65]. It has provided an algorithm for the recommended use and interpretation of hs-cTn I levels. Patients with hs-cTn I levels > 99th percentile at 0 h with >50% delta change at 3 h are classified as high-risk while those with 0 h hs-TnI > 99th percentile and <50% delta change are intermediate risk and assessed for alternate diagnoses; hs-TnI < 99th percentile at 0 h and <50% delta change are low risk [65]. 

Despite all these guidelines, clinicians should also be aware of the analytical aspects related to hs-cTn assays with respect to the sex-specific 99th-percentile values and the concept of delta change to facilitate their interpretation of results. This highlights the need for closer laboratory–clinician interaction and communication.

In contrast, some groups opine that troponin tests are widely overused even when there is a low possibility of AMI [70]. Only a minority of patients presenting with chest pain are ultimately diagnosed with AMI [26], while those who are late presenters (symptom onset > 3 h) are at a low risk of having any major adverse events [25,71,72]. As these patients can be identified on the basis of a single low hs-cTn measurement, serial troponin testing may not provide any substantial benefit. Moreover, it has been suggested that administering a single troponin test is safe when it is used in conjunction with clinical assessment and the use of risk scores such as HEART (history, ECG, age, risk factors, troponin) [73]. However, its wider adoption may depend on the clinician’s familiarity with these scores and the ease with which they can be incorporated into the local clinical pathways. Though the single-test approach seems reasonable in those with a low-risk presentation, evidence suggests that it might not be sensitive enough for patients presenting < 3 h from symptom onset; serial testing is recommended in this group [74]. In addition, many emergency department (ED) physicians already use serial testing to be reassured that the patients they discharge are low risk. This reduces the chances of a missing a diagnosis of AMI and minimizing the risk of major adverse cardiac events in patients soon after discharge [75].

### 3.3. Which Troponin: cTnT or cTnI?

Recent evidence suggests clinical and prognostic discrepancies between cTnT and cTnI, which can be attributed to biochemical differences between these troponins, their clearance mechanisms, and assay-related issues affecting the measured concentrations [50]. Notwithstanding these considerations, both hs-cTnT and hs-cTnI have consistently been associated with subclinical or manifest cardiovascular diseases and are deemed equally effective for clinical application [6,26]. Hs-cTnT has been suggested to be a stronger indicator of all-cause mortality while hs-cTnI appears to be more sensitive to coronary artery disease and ischemia [50]. Some variations also exist between the release patterns of hs-cTnT and hs-cTnI. In cases of MI as well as reversible ischemia, hs-cTnI concentrations increase faster compared to hs-cTnT, which might be due to a tighter bond between cTnT and tropomyosin, leading to a slower release of cTnT compared to cTnI.

When interpreting hs-cTnT concentrations, a stronger association with renal dysfunction and diabetes compared to hs-cTnI needs to be recognized [36]. Additionally, skeletal muscle expression of cTnT (but not cTnI) in patients with myopathies may also lead to discrepant results [76]. However, despite these discrepancies, the predictive differences between both troponins are not significant in real-life scenarios. 

The use of other biomarkers besides hs-cTn is not recommended unless hs-cTn is unavailable [26]. Myosin-binding protein C, CK-MB, and copeptin may have a complementary role when used with standard cTn.

### 3.4. Monitoring the Turnaround Time

The Laboratory Medicine and Practice Guidelines have recommended reporting the hs-cTn results within 60 min or less of when a sample is received by the lab [52]. In this regard, the Roche 9 min TnT assay [77] seems preferable over other platforms with longer assay times. Continued collaborative efforts should be taken to improve this to a time of 60 min from sample collection. This could be achieved by introducing dedicated pathways and protocols for hs-cTn analysis. Given the logistical constraints in obtaining a rapid turnaround time, it is not surprising to find many real-world applications of the 0/1 h rapid testing algorithm mentioned above. This can only be achieved with on-site POC hs-cTn devices. There are three such hs-TnI devices [78] that have been introduced—Pathfast (LSI Medience), Triage (Quidel), and Atellica VTLi (Siemens). They provide results in under 20 min. Pathfast and Triage lack studies on their clinical performance in AMI rule-in and rule-out. The most promising is the Atellica VTLi, with its analytical performance rigorously assessed in plasma and whole blood [79]. The Atellica’s clinical performance has been derived and validated in a US cohort (whole blood, *n* = 1086) against the Abbott hsTnI and in Australia (plasma, *n* = 1486) against the Beckman hs-cTnI [80]. Using a hs-TnI of < 4 ng/L, the Atellica identified patients at low risk of MI, death (cardiac and all-cause) and unplanned revascularization at 30 days. However, Atellica is available in only a few markets.

In addition, clinicians should recognize the importance of proper sample collection while implementing hs-cTn assays to avoid issues such as hemolysis and other interfering substances, which may lead to erroneous results. Close communication between laboratory and clinicians is strongly encouraged to ensure the delivery of timely and accurate hs-cTn results.

## 4. Future Directions

The ability of hs-cTn assays to detect the low cardiac troponin concentrations present in healthy subjects provides an exciting opportunity for employing these tests in community-wide cardiovascular risk screening [81,82]. There is growing enthusiasm for the use of hs-cTn assays for cardiovascular risk stratification in the general population [83]. Hs-cTn might be a valuable tool to identify individuals who would benefit from earlier lifestyle changes or preventive medications to reduce their CVD risk [81]. Another consideration would be the incremental value hs-cTn provides when added to well-established clinical indicators. Interestingly, data from the prospective observational Nord-Trøndelag Health (HUNT) study (*n* = 9005) showed that the addition of hs-TnI to established cardiovascular risk prediction models led to a net reclassification improvement of 35% for the prediction of future cardiovascular events [84].

Numerous studies have demonstrated the predictive value of elevated hs-cTnT and hs-cTnI levels for future cardiovascular events in asymptomatic individuals, as summarized in recent meta-analyses [82,83,85,86]. An observational study involving 15,340 subjects aged 48 years at recruitment in the Scottish Heart Health Extended Cohort showed that the Abbott hs-cTnI (LoD 1.9, Male 99th-percentile URL 31.7, and female 99th-percentile URL 18.1 ng/L, respectively) was an independent predictor of cardiovascular events [87]; subjects with troponins in the upper quartile had 2.5-fold risk versus those in the lowest troponin quartile. In asymptomatic subjects the recommended cut-offs for cardiovascular risk using the Abbott hs-cTnI (ng/L) is—low (M < 6, F < 4), moderate (M 6–12, F 4–10), and high (M > 12, F > 10), respectively [81]. In the Age, Gene/Environment Susceptibility-Reykjavik Study (AGES-Reykjavik) older cohort (mean age 77 years) followed up for 10 years, hs-cTnI > 10.6 ng/L was found to be a useful marker for the prediction of major adverse cardiovascular events (MACE) and overall death in seemingly healthy individuals [88]. Furthermore, the Atherosclerosis Risk in Communities (ARIC) study on an elderly population (>66 years) demonstrated that elevated hs-cTnI in individuals without CVD history was independently associated with increased mortality and cardiovascular risk similar to those with a clinical CVD history [89]. The ARIC study also showed that hs-cTnI improved discrimination for the prediction of heart failure and cardiovascular mortality.

Similar studies have reported an association between hs-cTnT levels and the risk of adverse cardiovascular events, including heart failure and mortality. Among the subjects from the Dallas Heart Study, hs-cTnT levels were associated with incremental risk in those without CVD history [90]. In a Chinese population (*n* = 1325), elevated hs-cTnT was a major risk factor for major cardiovascular events and all-cause mortality [91]. 

Another possible application of hs-cTn assays is the identification of subclinical CVD akin to a biochemical stress test. In a previous study, an intense 2 h bicycle ride in a middle-aged individual resulted in elevated cTnI and cTnT levels [49]. This was followed by the discovery of severe coronary artery stenosis on coronary angiography which was treated with a stent. Subsequent exercise months later showed only minor cTn elevations which reverted post-exercise [49]. While studies are underway to determine the mechanism of post-exercise cTn elevation [48], further investigations are necessary to fully utilize hs-cTn as a stress test for CVD screening. 

There are also several challenges that need to be addressed before community screening for hs-cTn can be widely implemented. One challenge is the cost of hs-cTn testing. In addition, community screening programs would need to be designed to ensure that individuals with elevated hs-cTn levels receive appropriate follow-up care.

## 5. Conclusions

Timely and accurate diagnosis is critical in the management of AMI. Hs-cTn assays offer enhanced sensitivity, but their effective use requires standardized practices. This review has outlined laboratory considerations for hs-cTn, focusing on establishing reference limits and quality control procedures. Timely reporting of hs-cTn results is also highlighted. In the clinical context, hs-cTn measurements are crucial for ruling in or ruling out AMI. While there are some differing opinions on the extent of serial testing, its importance in risk stratification and prognostic assessment cannot be understated. The implementation of community screening using hs-cTn assays should be carefully studied to define the optimal target populations and preventive interventions to ensure their effectiveness in improving cardiovascular risk assessment and reducing adverse outcomes. This review hopes to promote cohesive understanding among professionals, facilitating the optimal utilization of hs-cTn assays in clinical settings, and ultimately improve patient care.

## Figures and Tables

**Figure 1 diagnostics-14-00087-f001:**
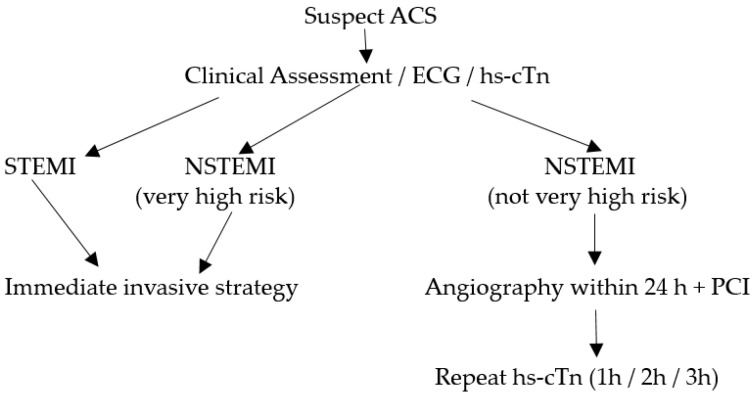
Algorithm for patients with suspected ACS.

**Figure 2 diagnostics-14-00087-f002:**
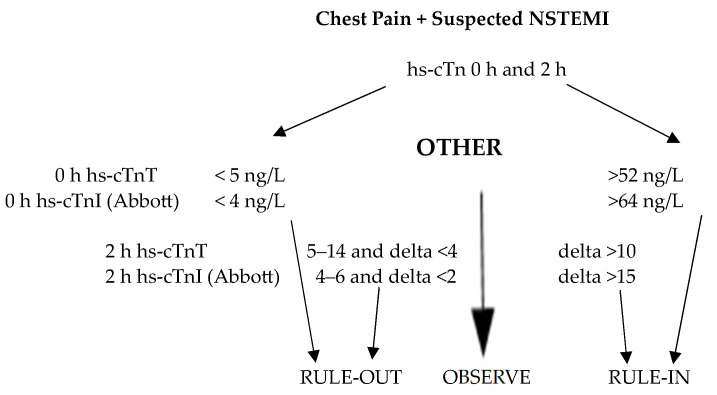
The 0/2 h rule-out/rule-in algorithm for suspected NSTEMI.

**Table 1 diagnostics-14-00087-t001:** Non-MI causes of elevated troponins.

Cardiac Conditions
Congestive cardiac failure (acute and chronic) [29]
Atrial fibrillation [43]
Stable coronary artery disease [28]
Pericarditis/myocarditis [44]
Aortic dissection [45]
Cardiotoxic chemotherapy [46]
**Non-cardiac conditions**
Chronic kidney disease [32]
Diabetes [32]
Sepsis [47]
Cancer [24]
Untreated thyroid disorders [39]
Strenuous exercise [48,49]
Presence of cTn autoantibodies [50]
Myopathies [50]
Ischemic stroke [51]

## Data Availability

Not applicable.

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
