# Peer review of "Optimizing the Clinical Use of High-Sensitivity Troponin Assays: A Review"

_diagnostics, 2023, doi:10.3390/diagnostics14010087_

Round 1

Reviewer 1 Report

Comments and Suggestions for Authors

The manuscript is a narrative review, focusing on cardiac troponins. It is a topic useful for clinicians. However, it needs important revision to fulfil its goal. 

Introduction

The introduction should also contain general information about troponins and their roles (diagnosis/prognosis/risk stratification).

Lines 29-30: cTn are important in the management of patients with acute chest pain, not for every case of IHD. Clinical suspicion for an acute coronary syndrome triggers the performance of an ECG. For STEMI patients the diagnosis and treatment do not require troponin assessment. For patients with suspicion of NSTEMI-ACS, the value of troponin is undeniable. This issue should be highlighted. For guidance, please find more data in the new ESC guideline for ACS (2023). doi: 10.1093/eurheartj/ehad191.

Lines 18-19 and 83-85 The 2023 ESC Guidelines state: “Therefore, until automated tools (i.e. risk assessment calculators) incorporating the effect of all four clinical variables (age, eGFR, time from chest pain onset, and sex) are available, the use of uniform cut-off concentrations should remain the standard of care for the early diagnosis of MI. This update should be integrated in the manuscript.  

Lines 179-181

Considering that there are differences recognized by the manufacturers and mentioned in the practice guidelines between the cut-off for cTn I and cTnT, and between hs assays and older assays, why would anyone want to compare them? In fact, this is one of the shortcomings of troponins: variability of the numerical value of the 99th percentile and of the delta value between assays (assay generation, troponin T or I, manufacturers).

Section 3.1. needs revision. Certain information is incomplete, which predisposes to misunderstandings.

Line 220: Elevated levels of high-sensitivity troponin I (comparative to what?)

Line 226: lower hs-cTn levels (comparative to what?)

Line 227-331: low elevations … mild hs-cTn elevations just above the 99th percentile…

It implies that low elevations are values still below 99th percentile, so there are normal values. Why call them elevations?

An expert panel from Asia-Pacific recommended that for ruling in AMI, hs-cTn levels >10 times the upper limit of normal should be considered. Recommendations from other scientific societies must be mentioned so that the entire spectrum of opinions is presented. It would be beneficial to create a table with the manufacturers' recommendations for cut-off and delta.

Lines 278-281. Only opinion of expert panel from Asia-Pacific is mentioned (0-3h algorithm). However, American and European guidelines recommend a different approach (0-1h/2h algorithm). This aspect should be discussed.

Section 3.4. The point-of –care possibility to determine hs-cTn with very low turnaround time should be highlighted. A table with the best available options may be useful (if possible).

Thank you!

Comments on the Quality of English Language

Minor editing of English language required.

Author Response

Thanks for your comments. Please check the file below.

Reviewer 2 Report

Comments and Suggestions for Authors

Tiwari et al. reviewed the laboratory and clinical considerations in the article titled "Optimizing the clinical use of high-sensitivity troponin assays: a review." The authors highlighted the importance of reference population (sex, diabetes, and renal dysfunction), derivation of the 99th percentile, and quality control. The authors also highlighted the importance of clinical utility and common pitfalls in troponin utilization in clinical practice.
Major concerns: None
Minor concerns: None

Author Response

Thanks for your precious comments.

Reviewer 3 Report

Comments and Suggestions for Authors

The authors aimed to review how to optimize the use of high-sensitivity troponin assays in clinical practice. However, the main aim of this review was not supported by the content of the manuscript. Although decently written, this review doesn’t bring any novelties to an already “classic” topic. The use of hs-TnI  is currently highlighted in the latest ESC Guidelines for Acute Coronary Syndromes (2023), the specific cut-offs and dedicated assessment strategies being also underlined (and based on multicenter studies).

The authors only mention the non-myocardial infarction conditions that alter the troponin levels, however, they did not detail the mechanisms by which this occurs (the rise of troponins in certain conditions).

The current paradigm is to focus on novel cardiac biomarkers or an integrative multimarker approach. The authors did not mention any of these cardiac biomarkers compared to hsTni.

The manuscript basically refers only to well-established, well-known information, without any novelties.

The review totally lacks figures or adequate graphic representations.

Best regards,

Comments on the Quality of English Language

Fine

Author Response

(The authors gave the same response as above.)

Round 2

Reviewer 1 Report

Comments and Suggestions for Authors

The Authors have revised the manuscript carefully and significantly improved its quality. They clarified all my concerns.

Minor:

Line 39-42. The Authors added new text. However, the statement is only partially correct and it should be revised. Recommended concept: Immediate reperfusion therapy is indicated in patients with STEMI and immediate angiography ± PCI is recommended in patients with very high-risk NSTE-ACS. Hs-cTn measurements should not delay the implementation of these measures. Thus, in the above-mentioned patients, troponin is important for its prognostic value, correlating with in-hospital and long-term mortality. In the remaining patients with suspected ACS, troponin is vital in diagnosis, risk stratification and subsequent case management. (https://doi.org/10.1093/eurheartj/ehad191)

Line 59: learned societies such as… Recommendation: reputable scientific societies such as

Line 181:  h-cTn (correct hs-cTn)

Thank you!

Comments on the Quality of English Language

No significant concerns. 

Author Response

Thanks for your comments. Our reply is below.

Reviewer 3 Report

Comments and Suggestions for Authors

First of all, I think that the approach of my review is not appropriate. My task here is to critically review the manuscripts, not to praise them. And the expected outcome is an improvement of the manuscript, not just to express explanations for the lack of some aspects.

- The ESC 2023 guidelines for acute coronary syndromes have been available online since 25 August 2023, right after its official presentation. The paper based on these guidelines was published on EHJ on October 7. It is of paramount importance to publish state-of-the-art information, the readers aim for the latest information, apart from any manuscript’s submission date.

So, as not to have doubts, I provided the official link to the guidelines where the specific timeline is mentioned:

https://academic.oup.com/eurheartj/article/44/38/3720/7243210?login=false

- Concerning the ESC recommendations (Section 3.3.5, page 3742- ESC 2023 guidelines ACS): “The use of biomarkers other than cTn for the diagnosis of ACS is not recommended (unless cTn is not available). Among the multitude of additional biomarkers evaluated for the diagnosis of NSTEMI, only creatine kinase myocardial band isoenzyme, myosin-binding protein C, and copeptin may have clinical relevance when used in combination with (standard) cTn T/I, although in most clinical situations their incremental value above and beyond cTn is limited.”

à I have recommended mentioning other biomarkers as an integrative approach, as the guidelines stated (see above).

- “We disagree that much of the troponin information is well known (maybe for the few afficionados).” R: It is an opinion, but it is not the scope of the Journal to assess the prevalence of expertise of the eventual readers in the area of troponin. I evaluated the novelty expressed by this manuscript from the perspective of a specialist who interacts with it on a routine basis.

- On one hand, in the first paragraph, you claim that you didn’t have the opportunity to timely read ESC guidelines, on the other hand, you address my observation concerning the graphic representation (figures) by sending the potential readers to the ESC guidelines. I could also claim that not all are “aficionados” who will eventually search for figures in the specific guidelines. Moreover, it is strictly about this manuscript, its graphical appearance, and the “readers-friendly” concept. It is awkward and non-academical to send readers to search for specific graphics in other authors' research work/reviews.

Comments on the Quality of English Language

fine

Author Response

(The authors gave the same response as above.)

Round 3

Reviewer 3 Report

Comments and Suggestions for Authors

I highly appreciate the author's efforts in addressing my recommendations. The quality and soundness of the manuscript has improved.